# Cryo-EM structures of inhibitory antibodies complexed with arginase 1 provide insight into mechanism of action

Rachel L. Palte [1✉], Veronica Juan[2], Yacob Gomez-Llorente[3], Marc Andre Bailly[2], Kalyan Chakravarthy[4,8], Xun Chen[3], Daniel Cipriano[2], Ghassan N. Fayad[5], Laurence Fayadat-Dilman[2], Symon Gathiaka[1], Heiko Greb[2,9], Brian Hall[6], Mas Handa[2], Mark Hsieh[2], Esther Kofman[2], Heping Lin[6], J. Richard Miller[4], Nhung Nguyen[2], Jennifer O'Neil[7,10], Hussam Shaheen[6,11], Eric Sterner[6], Corey Strickland[3], Angie Sun[6], Shane Taremi[6] & Giovanna Scapin[3,12]

Human Arginase 1 (hArg1) is a metalloenzyme that catalyzes the hydrolysis of L-arginine to L-ornithine and urea, and modulates T-cell-mediated immune response. Arginase-targeted therapies have been pursued across several disease areas including immunology, oncology, nervous system dysfunction, and cardiovascular dysfunction and diseases. Currently, all published hArg1 inhibitors are small molecules usually less than 350 Da in size. Here we report the cryo-electron microscopy structures of potent and inhibitory anti-hArg antibodies bound to hArg1 which form distinct macromolecular complexes that are greater than 650 kDa. With local resolutions of 3.5 Å or better we unambiguously mapped epitopes and paratopes for all five antibodies and determined that the antibodies act through orthosteric and allosteric mechanisms. These hArg1:antibody complexes present an alternative mechanism to inhibit hArg1 activity and highlight the ability to utilize antibodies as probes in the discovery and development of peptide and small molecule inhibitors for enzymes in general.

[1] Department of Discovery Chemistry, Merck & Co., Inc., Boston, MA, USA. [2] Department of Discovery Biologics, Merck & Co., Inc., South San Francisco, CA, USA. [3] Department of Discovery Chemistry, Merck & Co., Inc., Kenilworth, NJ, USA. [4] Department of Discovery Biology, Merck & Co., Inc., Boston, MA, USA. [5] Department of Preclinical Development, Merck & Co., Inc., Boston, MA, USA. [6] Department of Discovery Biologics, Merck & Co., Inc., Boston, MA, USA. [7] Department of Discovery Oncology, Merck & Co., Inc., Boston, MA, USA. [8] Present address: Ipsen Bioscience Inc., Cambridge, MA, USA. [9] Present address: Synthekine Inc., Menlo Park, CA, USA. [10] Present address: Xilio Therapeutics, Waltham, MA, USA. [11] Present address: Pandion Therapeutics, Cambridge, MA, USA. [12] Present address: NanoImaging Services, Woburn, MA, USA. ✉email: rachel.kubiak@merck.com

Human Arginase 1 (hArg1) is a critical endogenous regulator of the immune system and a key player in T-cell function. This enzyme is constitutively expressed by myeloid-derived suppressor cells (MDSCs), which are known immune system regulators. MDSCs have emerged as a key mediator of immunosuppression in human T-cell biology leading to significant decreases in the induction of antitumor activity[1–4]. hArg1 catalyzes the degradation of the conditionally essential amino acid L-arginine into L-ornithine and urea in the final step of the urea cycle[4–6]. This enzyme is present both intracellularly and excreted into the extracellular environment in a paracrine manner, with extracellular hArg1 maintaining its ability to deplete L-arginine[7–10]. As T-cells are dependent on L-arginine for growth and proliferation, its depletion leads to the effective suppression of T-cell immune responses and consequently supports the proliferation of tumor cells both in vitro and in vivo[4,6]. Activation of lymphocytes, specifically T-cells, via therapeutics targeted at immune checkpoint molecules, enhances tumor cell killing and has led to long-lasting responses across various cancers[11]. Support of hArg1 as an immuno-oncology target is found in research reports over the past decades where high levels of hArg1 activity are correlated with various types of cancer[6,12].

hArg1 is a trimeric metalloenzyme in which each monomer is ~35 kDa in size with an extended, narrow active site ~15 Å deep that is terminated by two catalytic manganese (Mn) ions 3.3 Å apart[13]. Several residues within the active site are critical for bridging the two Mn ions and in binding L-arginine[13,14], and have been the main target of small molecule inhibitors[15–21] Prior efforts to discover pharmacological agents to inhibit hArg1 have been focused on amino acid-derived small molecules of usually <350 Da[7] that are able to enter and bind to residues within the hArg1[14] active site. One avenue not previously described in literature for hArg1 inhibition is the use of therapeutic antibodies.

Monoclonal antibodies (mAbs) both in monotherapy and in combination regimens has emerged as one of the fastest growing and most effective therapeutic strategies for the treatment of solid tumors and hematological diseases. Between 2015 and 2017, the US FDA approved 27 therapeutic mAbs[22] increasing the total of clinically used mAbs and biosimilars in 2017 to 57 and 11, respectively[23]. As of late 2019, numerous companies were supporting over 550 novel antibody therapeutics in early phase clinical trials, with approximately half of these against oncology targets[24].

Published studies have focused on the use of antibody fragments (Fabs) such as nanobodies, antigen-binding Fabs, and single-chain variable domain fragments as potent inhibitors of enzyme activity[25–34]. The proposed mechanisms of inhibition by antibodies include adaptation to the catalytic site; adaptation to a site other than but near to the catalytic center thereby causing steric hindrance; aggregation of the antigen-antibody complex leading to steric hindrance by the structure of the aggregate; and interference with multimerization that may inhibit enzyme activity[35]. Nevertheless, as noted by others, despite the myriad of antibodies that have been and could be developed, the number of full-length mAbs acting as enzyme inhibitors is "disappointingly low"[28]. MAbs excel in their ability to bind an antigen with high specificity and potency and function mainly by binding to large, flat surfaces, both on receptors and protein:protein interaction surfaces to which traditional small molecules cannot bind with suitable potency. However, full-length neutralizing antibodies often lack the ability to access the narrow clefts and active site pockets of traditional enzymes due to their larger size, which often eliminates the ability to inhibit enzymatic activity.

Despite hArg1 having a very narrow active site channel most suited for small molecule inhibitors, the presence of hArg1 in the extracellular space, and the extensive accessible surface area of trimeric hArg1, opens the possibility that antibodies could effectively inhibit hArg1 enzymatic activity. Indeed, we successfully identified and characterized five potent full-length anti-hArg1 antibodies. Enabled by cryo-electron microscopy (cryo-EM) studies, we present here the full structural analysis of five large hArg1:mAb macromolecular complexes, each of which is composed of trimeric hArg1 bound to potent anti-hArg1 antibodies. These mAbs differ both in epitopes and constant domain backbones, leading to varying complex structures. The availability of these high-resolution three-dimensional complexes allows for elucidation of the structural details of the antibody-antigen complex and facilitates the design of future inhibitory antibody variants against hArg1. In addition, the insights on steric occlusion of enzyme active sites is applicable to numerous enzymatic proteins and could serve as a framework for targeting other enzymes present in the extracellular compartment. Specifically, using antibodies as probes for new means of inhibition assists in the expansion of inhibitor chemical diversity.

## Results

Five mAbs (labeled mAb1 through mAb5) from three separate antibody generation campaigns are presented here. In general, all antibodies are bound to the hArg1 trimers in large, asymmetric complexes with most complexes being composed of two hArg1 trimers and three mAbs (referred to as a 2:3 complex; simplified depiction in Fig. 1a). This 2:3 ratio was initially assigned using both isothermal titration calorimetry (ITC) (Supplementary Information Fig. 1) and size exclusion chromatography with multi-angle light scattering (SECMALS) (Supplementary Information Fig. 2). Cryo-EM analysis reveals that the antigen-binding Fabs of the mAbs bridge the hArg trimers with the Fc region nearly perpendicular to the Fab arms. One Fab of each mAb binds to the "top" hArg1 trimer while the other Fab binds to the "bottom" hArg1 trimer. The Fc (Fragment, crystallizable) regions of the antibodies do not interact with the hArg1 trimers and are only visible in unfiltered, unflattened maps at a very low contour (Fig. 1b). While the full mAbs were indeed part of the visualized complexes we have chosen not to show the Fc regions in the finalized structures due to the inability for clear placement and interpretation at an atomic level. Figures showing a representative micrograph, 2D classes, a flow chart of the main processing steps, and Gold-Standard Fourier-shell correlation curves for all structures can be found in Supplementary Information Figs. 3–5.

In all cases, the resolution of the maps is highly variable across the complex with the local resolution at the hArg1 trimer:Fab variable domain interface at 3.5 Å or better (unsharpened and sharpened maps Fig. 1c and 1b). This allows for unambiguous tracing of the loops, assignment of the side chains, and determination of the epitope-paratope interactions within each complex.

The 2:3 complexes are asymmetric such that the one half of the complex, consisting of one hArg1 trimer and three Fabs, is slightly rotated with respect to the other half. Within each 2:3 complex, the angle of each of the three mAb backbones is slightly different and therefore no perfect twofold symmetry could be applied between top and bottom halves of the complex during map reconstruction. This knowledge, in addition to the fact that the interactions between hArg1 and mAbs1-4 are identical for both halves, led to refinements focused on one-half of the complexes. This allowed for an increase in resolution of one half at the expense of resolution on the second half. C3 symmetry can be applied within each half of the complex and was used in the masked map reconstructions to get atomic details for the Fab/hArg1 interfaces. All of the following results and discussions regarding epitope:paratope interactions for mAb1 to mAb4 will

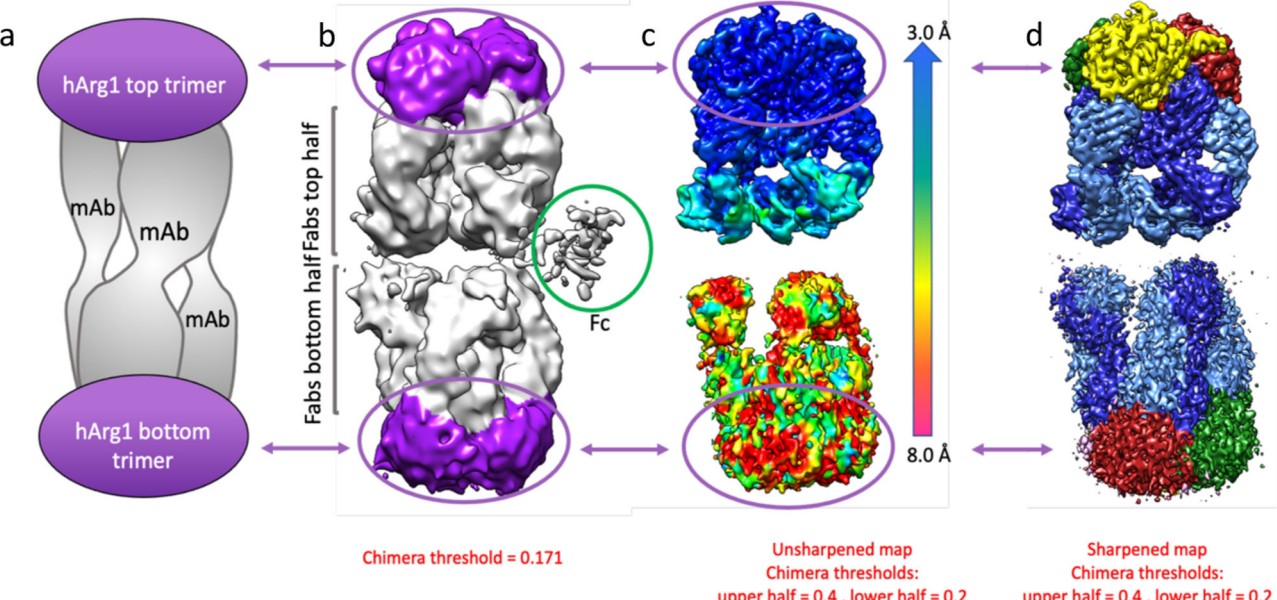

**Fig. 1 Simple depiction and cryo-electron microscopy maps of the 2:3 hArg1:mAb1 complex. a** Most complexes presented in this paper consist of two hArg1 trimers on the top and bottom of the complex spanned by three full mAbs, giving the complex a sandwich-like appearance. **b** An exemplar map for the hArg1:antibody structures described in this paper, colored to match the first image. **c** A color shell based on local resolution is overlaid with an unsharpened map. It is apparent that the more highly-resolved top half of the complex is almost completely within the 3–4 Å range. It is likely that the bottom half is just as ordered, but its flexibility relative to the top half results in the seeming loss of resolution. The top half of each complex was used to map epitope and paratope interactions. **d** A sharpened map of the complex.

be in reference to only half of the complex. The complex seen with hArg:mAb5 differs and will be discussed separately. Cryo-EM data collection, refinement, and validation statistics for all analyzed structures are shown below in Table 1 and Table 2. A comparison of the size and shape of the macromolecular complexes is discussed in futher detail in Supplementary Note 1 and Supplementary Information Fig. 8.

Each antibody was assessed for its potency and extent of inhibition of human (or mouse) arginase using an assay that detects formation of thioornithine from thioarginine. In this assay, as well as an LCMS assay, detecting the production of from L-arginine, it is determined that all antibodies could fully inhibit the enzymatic activity of arginase. To assess the kinetic mechanism of inhibition of each antibody, increasing fixed concentrations of antibody were tested across a range of thioarginine concentrations. Data were fit to various models of inhibition including competitive, mixed, noncompetitive, and uncompetitive, and the quality of fit was evaluated using the Aikakie information criteria[36]. (Supplementary Information Fig. 6). In each case the inhibition data was best fit by a competitive inhibition model and the fitted data are shown in Supplementary Information Fig. 6. Inferior fits of the data to other models of inhibition are not shown. The Ki for mAb1 and mAb2 against human Arg1 was $3.3 \pm 0.3$ nM and $5.3 \pm 0.8$ nM respectively. The $K$i of mAb5 against mouse Arg1 was $25 \pm 1.7$ nM (Supplementary Information Fig. 7).

**mAb1, mAb2, and mAb3 2:3 complexes**. MAb1, mAb2, and mAb3 are human antibodies identified via yeast display technologies. MAb1 is constructed on a mouse IgG2a/kappa backbone with a fairly flexible hinge region. MAb2 is constructed on a mouse IgG1 D265A/kappa backbone, which is more rigid and more conformationally restricted than that of mAb1. Both antibodies share identical variable regions, differing only in their

constant domains. MAb3 is constructed on a human IgG4 S228P/kappa backbone and is an affinity-matured version of mAb1, differing by eight amino acids in the heavy chain (HC) variable domain, with identical light chain (LC) variable domains. Despite these differences, all three mAbs share the same epitope, leading to identical interactions with hArg1. The overall characterization for these three 2:3 complexes are therefore presented in tandem.

Figure 2a shows the overall arrangement of the 2:3 complex consisting of two hArg1 trimers and the three mAbs. The hinge region of each mAb1-3 backbone holds an approximate T-shape with an angle of ~150° whereby each Fab from a single mAb interacts with only one of the two trimers. While all three Fcs are present, for clarity only a single one is displayed in the image.

Due to the locally applied C3 symmetry, the interactions between hArg1 and mAb are equivalent around the trimer. For the following analysis, the monomers of each trimer are labeled as A, B, and C as seen in Fig. 2b and Fig. 2c. Each Fab interaction spans the interface of two monomers of the trimer, such that Fab1 binds to both monomers A and B in the same hArg1 trimer, Fab1' binds to monomers B and C, and Fab1" binds to monomers C and A (Fig. 2d). The HC of each Fab interacts with two monomers, while the LC interacts with only one. The buried surface area between mAb1 HC + LC and one monomer of hArg1 is 1020 Å², while the surface area between mAb1 HC and a second monomer of hArg1 is only 372 Å² (Supplementary Information Table 2). The rest of the complex follows this pattern with each monomer having two sets of interactions: binding to the HC and LC of one Fab and only the HC of a second Fab. A closeup view of one hArg1:mAb1 interface is shown in Fig. 2d as a representative image of the numerous surface interactions found between hArg1 and mAbs.

The local resolution of ~3 Å and high-quality electron density map (Fig. 2e) permitted the determination of the epitope and paratope interactions between mAbs1-3 and hArg1. MAbs1-3 have prolonged HC CDR-3 loops that extend toward the opening

**Table 1 Cryo-EM data collection, refinement and validation statistics for mAb1 and mAb2 structures.**

| | mAb1 full complex 2 trimers:3 mAbs (EMDB-23293) (PDB 7LEX) | mAb1 Masked (−) (−) | mAb1 2 trimers:2 mAbs (EMDB-23295) (PDB 7LEZ) | mAb2 full complex 2 trimers:3 mAbs (EMDB-23296) (PDB 7LF0) |
|---|---|---|---|---|
| *Data collection and processing* | | | | |
| Magnification | 130,000 | 130,000 | 130,000 | 130,000 |
| Voltage (kV) | 300 | 300 | 300 | 300 |
| Electron exposure (e–/Å$^2$) | 45.46 | 45.46 | 45.46 | 44.32 |
| Defocus range (μm) | −1.0 to −1.8 | −1.0 to −1.8 | −1.0 to −1.8 | −1.2 to −2.0 |
| Pixel size (Å) | 1.04 | 1.04 | 1.04 | 1.04 |
| Symmetry imposed | C3 | C3 | C1 | C3 |
| Initial particle images (no.) | 94,796 | 94,796 | 94,796 | 52,550 |
| Final particle images (no.) | 37,385 | 77,986 | 16,810 | 52,550 |
| Map resolution (Å) | 3.6 | 3.4 | 4.1 | 3.7 |
| FSC threshold | 0.143 | 0.143 | 0.143 | 0.143 |
| *Refinement* | | | | |
| Initial model used (PDB code) | 6V7C | 6V7C | 6V7C | 6V7C |
| Model resolution (Å) | 3.6 | 3.5 | 4.3 | 3.8 |
| FSC threshold | | 0.5 | 0.5 | 0.5 |
| Map sharpening *B* factor (Å$^2$) | 106.7 | 134.6 | 77.9 | 109.7 |
| Model composition | | | | |
| Non-hydrogen atoms | 25,662 | 25,662 | 21,628 | 25,797 |
| Protein residues | 4554 | 4554 | 3654 | 4542 |
| Ligands | 0 | 0 | 0 | 0 |
| *B* factors (Å$^2$) | | | | |
| Protein | 76.2 | 118.8 | 224.2 | 56.6 |
| Ligand | — | — | — | — |
| R.m.s. deviations | | | | |
| Bond lengths (Å) | 0.009 | 0.011 | 0.007 | 0.010 |
| Bond angles (°) | 1.00 | 1.09 | 1.14 | 1.07 |
| Validation | | | | |
| MolProbity score | 1.89 | 1.96 | 2.14 | 2.03 |
| Clashscore | 6.42 | 6.9 | 11.4 | 7.5 |
| Poor rotamers (%) | 0.31 | 0.78 | 0.65 | 1.09 |
| Ramachandran plot | | | | |
| Favored (%) | 90.4 | 89.3 | 89.2 | 87.8 |
| Allowed (%) | 8.7 | 9.8 | 10.1 | 11.4 |
| Disallowed (%) | 0.89 | 0.89 | 0.66 | 0.80 |

For mAb1, the full complex was not refined in Phenix. For the other complexes, the relatively low values for the CC Mask, Volume and Peaks observed for the refinement of the full complex reflect the high degree of variability in the resolution range.

of the hArg1 active site (Fig. 2f). Inactivation of hArg1 is achieved by the presence of Tyr104 that inserts into and sterically blocks the hArg1 active site near the exterior surface of the enzyme (Fig. 2f). Overall steric occlusion of the active site entry channel appears responsible for inhibition of hArg1. Supplementary Information Table 3 summarizes the specific epitope-paratope interactions between hArg1 and mAbs1-3.

**mAb1 2:2 complex**. Interestingly, two classes of hArg1:mAb1 complexes that were identified during 3D classification were not present in mAb2 or mAb3 samples. Approximately 80% of all complexes identified were composed of two hArg1 trimers and three mAb1s, forming a 2:3 complex, while the remaining 20% of complexes were composed of two hArg1 trimers and two mAb1s (hereafter called a 2:2 complex). Although the resolution of the map was lower (3.6 Å for 2:3 versus 4.1 Å for 2:2), the high-quality map (Fig. 3a) allowed for both the two hArg1 trimers and two mAbs to be manually positioned. In this smaller complex, the angle between the Fabs is only ~60° and results in the two hArg1 trimers being positioned significantly closer together (Fig. 3b). The epitope and paratope interactions between each Fab and hArg1 monomers are identical to the 2:3 complex, however one set of interactions is absent due to having only two mAb1s present. (Fig. 3c). Here, Fab1 binds to both monomer A and monomer B, Fab1′ binds to monomers B and C, but the Fab that would bind across the monomer C and A is absent. Here, both monomer B and monomer C are inhibited by the HC CDR-3 loops of Fab1 and Fab1′, respectively, while the active site of monomer A remains open and uninhibited. Surface representations of this complex can be found in Fig. 3d. More discussion on the 2:2 complex can be found in the Supplemental Information (Supplementary note 2).

**mAb4 2:3 complex**. A mouse antibody campaign led to the discovery of mAb4, which was humanized and constructed on a human IgG4 S228P/kappa backbone. The hArg1:mAb4 interactions led to a similar 2:3 complex as seen with mAbs1-3 (Fig. 4a). However, the HC of mAb4 interacts with only one of the hArg1 molecules (monomer A in Fig. 4b), and the LC interacts with both monomer A and monomer B. In this complex, the mAb4 backbone is positioned so that the Fabs are at a ~120° angle. Since the hArg1:mAb4 complex is less linearized compared to complexes with mAbs1-3 the complex is visibly shorter and rounder than the 2:3 complexes described for mAbs1-3.

**Table 2 Cryo-EM data collection, refinement and validation statistics for mAb3, mAb4 and mAb5 structures.**

| | mAb3<br>2 trimers:3 mAbs<br>(EMDB-23297)<br>(PDB 7LF1) | mAb4-HiRes<br>(−)<br>(−) | mAb4—full<br>2 trimers:2 mAbs<br>(EMDB-23298)<br>(PDB 7LF2) | mAb5<br>1 trimer:3 mAbs<br>(EMDB-23294)<br>(PDB 7LEY) |
|---|---|---|---|---|
| *Data collection and processing* | | | | |
| Magnification | 130,000 | 130,000 | 130,000 | 130,000 |
| Voltage (kV) | 300 | 300 | 300 | 300 |
| Electron exposure (e–/Å$^2$) | 44.47 | 44.47 | 44.47 | 45.44 |
| Defocus range (μm) | −1.0 to −2.0 | −1.0 to −2.0 | −1.0 to −2.0 | −1.0 to −2.0 |
| Pixel size (Å) | 1.04 | 1.04 | 1.04 | 1.04 |
| Symmetry imposed | C3 | C3 | C3 | C3 |
| Initial particle images (no.) | 87,069 | 41,176 | 41,176 | 80,738 |
| Final particle images (no.) | 52,241 | 31,797 | 31,797 | 44,345 |
| Map resolution (Å) | 4.0 | 3.5 | 3.7 | 3.1 |
| FSC threshold | 0.143 | 0.143 | 0.143 | 0.143 |
| *Refinement* | | | | |
| Initial model used (PDB code) | 6V7C | 6V7C | 6V7C | 6V7C |
| Model resolution (Å) | 4.3 | 3.8 | 4.0 | 3.2 |
| FSC threshold | 0.5 | 0.5 | 0.5 | 0.5 |
| Map sharpening *B* factor (Å$^2$) | 131.8 | 117.9 | 108.4 | 89.0 |
| Model composition | | | | |
| Non-hydrogen atoms | 34,517 | 33,798 | 33,798 | 2265 |
| Protein residues | 4548 | 4452 | 4452 | 4902 |
| Ligands | 0 | 0 | 0 | 0 |
| *B* factors (Å$^2$) | | | | |
| Protein | 299 | 149 | 363 | 73.6 |
| Ligand | — | — | — | — |
| R.m.s. deviations | | | | |
| Bond lengths (Å) | 0.006 | 0.011 | 0.007 | 0.010 |
| Bond angles (°) | 1.05 | 1.09 | 1.14 | 1.07 |
| Validation | | | | |
| MolProbity score | 2.20 | 2.04 | 2.15 | 2.24 |
| Clashscore | 12.0 | 5.4 | 11.2 | 7.2 |
| Poor rotamers (%) | 0.16 | 0.68 | 0.16 | 0.79 |
| Ramachandran plot | | | | |
| Favored (%) | 86.7 | 88.4 | 89.0 | 87.9 |
| Allowed (%) | 12.9 | 11.2 | 10.9 | 11.4 |
| Disallowed (%) | 0.40 | 0.39 | 0.14 | 0.67 |

The relatively low values for the CC Mask, Volume and Peaks observed for the refinement of the full complex reflect the high degree of variability in the resolution range.

MAb4 binds directly over the opening to the active site, thereby sterically blocking the hArg1 active site similar to what is seen with mAbs 1-3. Here, Arg28 on the LC CDR-1 loop (orange in Fig. 4c) inserts into and occludes the active site of hArg1, resulting in total inhibition. In addition, Gln27 is positioned almost parallel to the surface of the active site channel, further sterically blocking substrate from entering. A complete list of the epitope and paratope interactions between hArg1 and mAb4 are summarized in Supplementary Information Table 4.

**mAb5.** Using a transgenic mouse platform, mAb5 is a human antibody constructed on a mouse IgG1 D265A/kappa backbone. In contrast to all previously described mAbs, the density for the second (i.e., bottom) half of the mAb5:hArg1 complex is almost completely absent, visible only in maps in which the top portion has been masked out and at very low contour (Fig. 5a). In this low-resolution portion of the map, the Fabs are oriented in a different conformation in which the termini of the three antibodies are positioned significantly closer to each other than in the previously described 2:3 structures. This suggests that no second hArg1 trimer is present and that the dominant form in the sample analyzed is one hArg1 trimer to three mAb5 antibodies (i.e., a 1:3 complex) (Fig. 5b). Lacking this second hArg1 trimer, the "bottom" set of Fabs are no longer conformationally restricted

through binding interactions with hArg1. This flexibility correlates with the absence of clear density for the entire second half of the complex. Distinct from previously-described mAbs, mAb5 binds to only a single hArg1 monomer, thereby sharing a relatively small interface with hArg1 that covers only 908 Å$^2$ (Supplementary Information Table 2).

The cryo-EM density at the enzyme:antibody interface of the top half was very good and allowed for a complete assignment of the main chain and side chains of both hArg1 and mAb5. Analysis of the epitope:paratope interactions reveal that the binding of mAb5 does not sterically block the hArg1 active site which remains accessible despite mAb5 being a neutralizing antibody. In addition, when compared to the other structures in this paper and to previously determined small molecule-bound structures[20] a small rearrangement of an hArg1 loop containing residues Lys16-Val24 is seen (Fig. 5d). The inhibition of hArg1 occurs through a different mechanism of action as discussed below in which mAb5 interacts with an amino acid important to substrate binding by hArg1. Supplementary Information Table 5 summarizes the residues involved in forming the epitope-paratope interactions between hArg1 and mAb5.

**Antibody binding affinities to trimeric hArg1 and monomeric hArg1.** Monomeric hArg1 was previously created by the

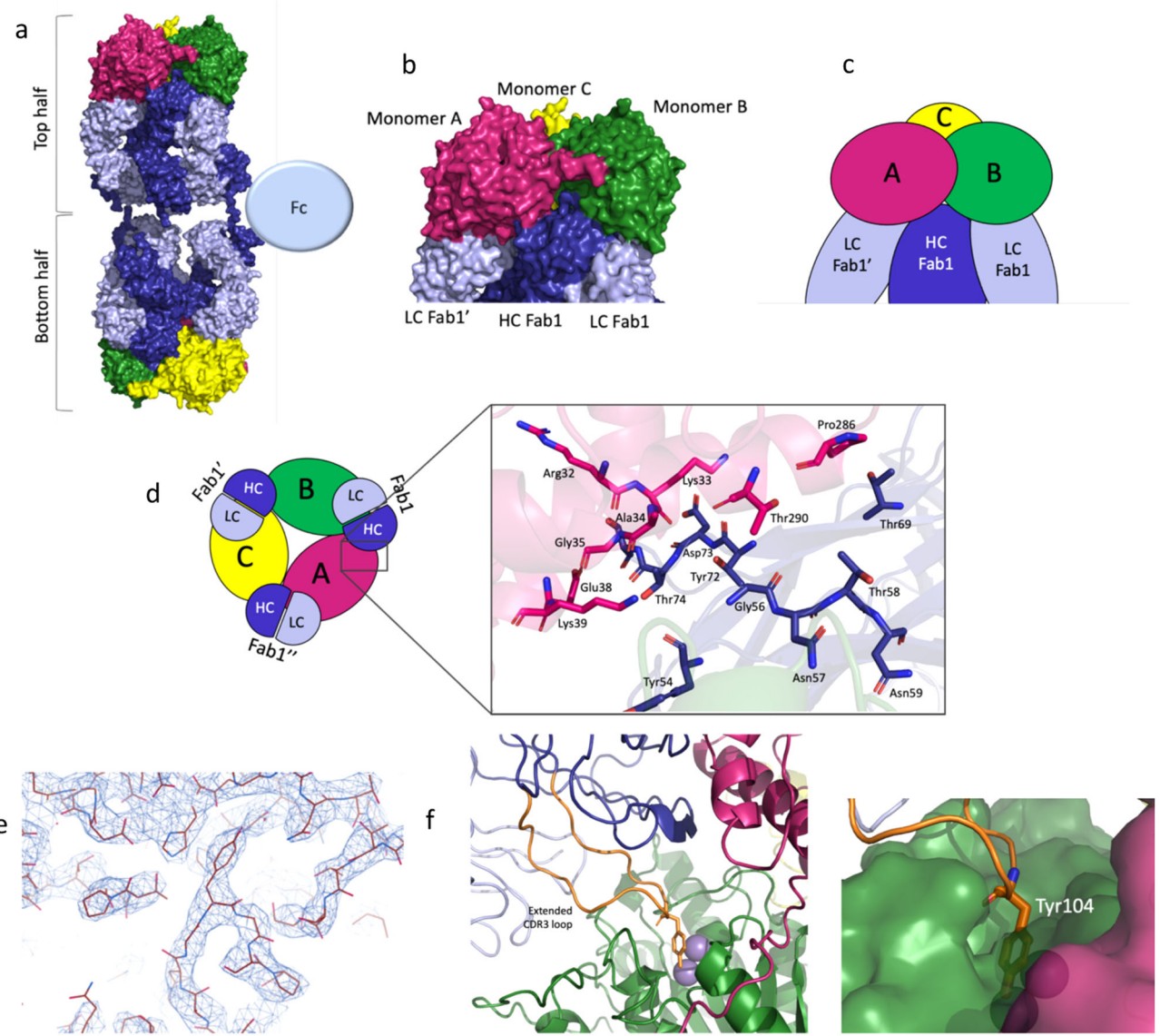

**Fig. 2 Epitope determination for mAb1, mAb2 and mAb3.** This panel shows an overview of how the large 2:3 hArg1:mAb1 complex assembles. The three monomers of the hArg1 trimer are colored in green, pink, and yellow, and the mAbs colored in dark blue (heavy chain) and light blue (light chain). **a** A surface representation of the full complexes. **b** A closeup view of mAb1's interaction across two hArg1 monomers in surface representation. **c** A closeup view of the mAb1:hArg monomers in simplified cartoon representation. **d** The complex is viewed in a simplified cartoon form as viewed from the top. Each HC interacts with two hArg1 monomers while each LC interacts with only one hArg1 monomer; these interactions are symmetric around the trimer. A closeup view of the monomerA:HC surface interaction is provided. **e** A sample of the electron density at the hArg1:Fab interface is shown, highlighting the ability to confidently model all main chain and side chain atoms. **f** This class of mAbs are characterized by a very long CDR-3 loop (orange). Tyr104 (shown as sticks) extends into the hArg1 active site. The binuclear active site manganese ions are shown as purple spheres.

mutagenesis of Arg308 found at the monomeric interfaces to an alanine residue[37]. Arg308 is important for maintaining a salt bridge between each hArg1 monomer and mutation of this residue leads to monomerization of hArg1 and nearly 85% loss in enzymatic activity[37]. We sought to explore the affinity changes of these antibodies when bound to monomeric hArg1 versus trimeric hArg1. The affinity measurements of mAb3 and mAb4 binding to both trimeric and monomeric hArg1 were determined by surface plasmon resosnance (SPR) studies to gain insight into how hArg1 oligomerization corresponds to antibody binding. MAb3's binding to trimeric hArg1 was quite potent (KD = 0.74 nM), but there was no measurable binding between mAb3 and monomeric hArg1 (Supplementary Information Table 6). The affinity of mAb4 for hArg1 differed only by ~36 fold between trimeric hArg1 (0.56 nM) versus monomeric hArg1 (20 nM)

(Supplementary Information Table 6). A more detailed discussion of this data can be found in the Supplemental Information (Supplementary note 3).

## Discussion

**Historical shortage of enzyme-inhibiting monoclonal antibodies.** There are numerous enzymes that are important therapeutic targets. It is often difficult to inhibit these enzymes selectively due to high similarities within the active site. Small molecules have been the canonical inhibitor of choice based on their relatively small size and ability to access the active sites. However, the use of small molecules as therapeutics faces significant challenges when it comes to targeting a single member of a family of considerably homologous enzymes. Here, antibodies

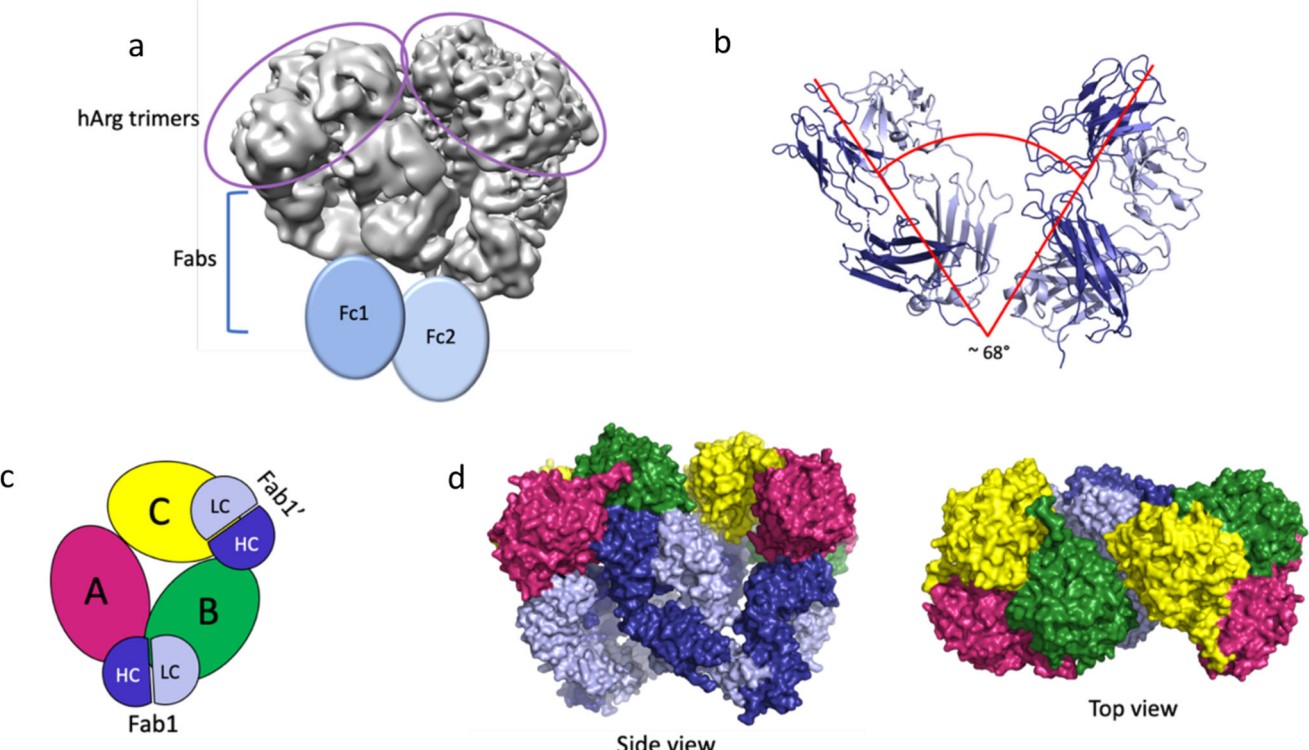

**Fig. 3 The mAb1 2:2 complex. a** The resulting EM maps allow for the identification of the two trimers and two sets of Fabs. As with the 2:3 complex, the Fcs were not visible but are added here to show their general location in this complex. **b** The angle between the two FABS is ~67° which is significantly smaller than the ~160° angle seen in the 2:3 complex. **c** An overview of how the 2:2 complex assembles with hArg1 monomers and the mAbs colored as in Fig. 2 is shown as a simplistic cartoon representation. Here it is apparent that that arrangement of the antibodies is quite asymmetric and only one monomer (monomer B here) makes all three mAb interactions (mAb1 LC and HC; and mAb2 HC) as seen in the 2:3 complex. **d** Two surface representations show both a side view and a top view of this 2:2 complex.

can play a vital role due to their ability to bind an antigen with high specificity and potency.

Enzyme active sites are nearly always situated within a pocket on the protein surface with the vast majority (>80%) of substrates binding within the largest cleft of the enzyme[38]. This often results in the inability of antibodies to directly contact active site residues as mAbs are bulky molecules that most commonly bind to large, flat surfaces and cannot access the enzyme catalytic site. The characteristics of the antigenic region on the antibody differs depending on the molecules to which it binds. With regards to binding locations on the antibody, more often than not proteins bind to a planar surface, haptens bind within a cavity-shaped pocket, and DNA, peptides, and carbohydrates bind within a groove-shaped pocket[39]. The structures presented here are crucial for defining different mechanisms by which multiple potent and specific antibodies inhibit hArg1 enzymatic activity. Several structures have been released in the PDB[40,41] showing antibody:protein interactions (ex: 3HFM, 4NZR, and 5N88 among others), and numerous studies—and structures such as 6SS2 and 6SS4[42]—have been completed using antibody Fabs[25–34]. However, wholly resolved X-ray or cryo-EM structures of full-length inhibitory mAbs bound to enzymatic proteins, as presented in this work, are absent. Other currently published structures are based on computational docking/posing[43], binding studies[44], competitive ELISA assays[45], mutagenesis studies such as alanine scanning[46], or low resolution Small Angle X-ray Scattering (SAXS) data[47], along with recently described inhibitory hArg2 mAbs for which Fab-bound hArg2 structures are available[48].

The large macromolecular complex structures solved in this current work allow for analysis of the specific epitope:paratope interactions between anti-hArg1 antibodies and hArg1. A structure with full-length mAbs also provides an opportunity to visualize and interpret a mechanism of action, which is constrained or even enabled by the avidity inherent in a mAb versus Fab. This work can be directly applied to facilitate the design of future antibody variants, and highlights the ability of utilizing structural-based methods to determine the inhibitory mechanism of mAbs against enzymatic targets.

**Mechanism of inhibition by the anti-hArg1 antibodies.** All five of the antibodies presented here are potent inhibitors of hArg1 enzymatic activity and were determined to have a competitive mechanism of inhibition. The means of inhibition by mAbs1-3 and mAb4 is based on steric occlusion of the hArg1 active site. The shared surface area between a single hArg1 trimer and three Fabs is ~4200 Å², and the interactions across these surfaces clearly contribute significantly to the overall potency of the mAbs. However, as shown in Fig. 6 and described previously, all four of these antibodies function by inserting an amino acid side chain into, and therefore sterically blocking, the narrow active site channel. This is accomplished by Tyr104 on the HC CDR-3 loop in mAbs1-3, and by Arg28 on the LC CDR-1 loop of mAb4, which occupy the same space in the overlaid structures. While mAb1, mAb2, and mAb3 utilize Tyr104, the subsequent Arg105 of these mAbs is also near the active site opening but does not access the narrow channel as seen with mAb4. The side chains of arginine and tyrosine are chemically different in size and charge, so it seems that, rather than relying on a specific residue for inhibition, overall steric occlusion of the active site is the inhibitory mechanism of the antibodies. This also hints at the ability to design small molecule inhibitors that are focused more on

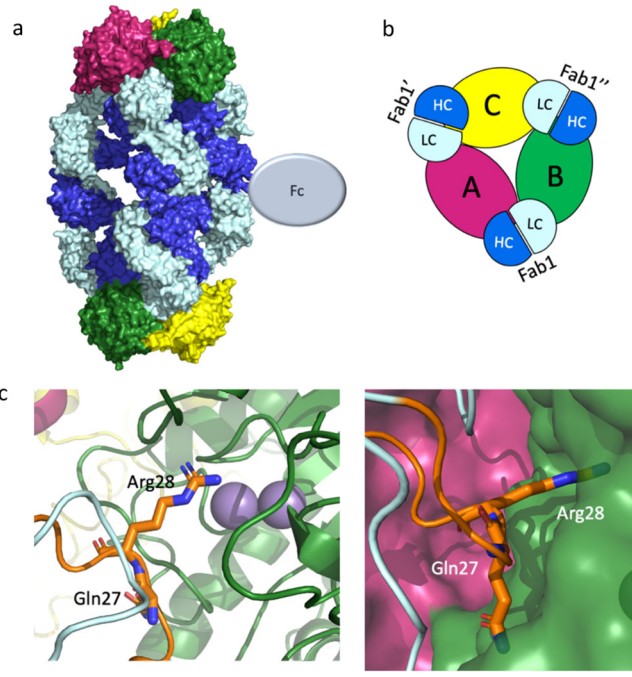

**Fig. 4 Overall complex formation and epitope mapping for mAb4 2:3 complex. a** An overview of the 2:3 hArg1:mAb4 complex assembly. The three monomers of the hArg1 trimer colored in green, pink, and yellow, and the mAbs colored in bright blue (heavy chain) and pale blue (light chain). The full complex is shown in surface representation. **b** Half of the complex in cartoon form is viewed from the top down for ease of interpretation. Here it is apparent that each HC interacts with only one hArg1 monomer, while each LC interacts with two hArg1 monomers. These interactions are symmetric around the trimer. **c** In this mAb, the CDR-1 loop of the LC (orange) is in close proximity to the hArg1 active site with Arg28 (sticks) extending into the binding pocket. Gln27 (sticks) occupies additional space on the outer surface of the active site. The binuclear active site manganese ions are shown as purple spheres.

surface-based steric occlusion instead of requiring full active site infiltration.

Unlike mAbs1-4, the inhibition of hArg1 by mAb5 is not due to obvious steric occlusion of the active site. The paratope is offset from the hArg1 active site and the active site remains accessible. The structure of mAb5 bound to hArg1 reveals a slight movement of an hArg1 loop containing residues Lys16-Val24 (Fig. 5d) as compared to hArg1:mAb1 and hArg1:small molecule[20] structures. Previous computational work on hArg1 has shown that when the active site opens to allow the entrance of substrate, residues Arg21 and Thr246, which sit on the outer lip of the active site entrance, are the main substrate interacting partners[37]. It was also determined that product movement toward the hArg1 cavity exit is stabilized through an interaction with Arg21[37]. In the hArg1:mAb5 structure, the loop containing Arg21 adopts an altered conformation in which residue Arg21 on hArg1 moves by ~6.1 Å (Fig. 7a). This is driven by Arg21 participating in a salt bridge with Asp30 of mAb5 (Fig. 7b). An in-house molecular dynamics simulation supports that Arg21 is strongly engaged by the antibody through a salt bridge with Asp30, suggesting it may not be available for binding to substrate and thereby inhibiting enzyme activity. This bond between the guanidino donor of Arg21 and the carboxylate acceptor of Asp30 had an average distance of 2.8 Å and an angle of 153°. The distance between the guanidino carbon atom (CZ of Arg21) and the carboxylate carbon (CG of Asp30) over the 100 ns simulation indicates that the salt

bridge is realized throughout the simulation time (Supplementary Information Fig. 9). These results confirm that the likely mechanism of inhibition of mAb5 is through tying up Arg21, preventing it from forming the necessary interactions with both substrate and product. As with mAbs 1–4, this is a surface-based mechanism of competitive inhibition that could be used a template for the design of future inhibitors.

**Impact of the 2:3 complex on hArg1 inhibition**. The fully discernable formation of the large macromolecular complexes consisting of two trimers and three full mAbs is unique in the literature. Recently, a similar complex of a single HtrA1 trimer and three anti-HtrA1 Fabs was determined through negative staining EM[47] analyses coupled with a previously determined low-resolution SAXS structure of the HtrA1 trimer[49] to build a 1:3 model of enzyme:antibody. The full "cage-like" structure was proposed and generated by juxtaposing and mirroring two of the HtrA1:Fab 1:3 complexes resulting in a 2:3 HtrA1:Ab complex much like those presented here. In vitro enzymatic assays determined that anti-HtrA1 full-length antibodies had greater than tenfold higher potency versus Fabs alone, hinting that the ability of HtrA1 to form these large macromolecular complexes of two HtrA1 trimers to three antibodies is instilling higher potency. While singular anti-hArg1 Fab arms were not tested in our studies, we hypothesize that all mAbs that showed distinct 2:3 complexes (here, mAbs1-4) are also highly potent due in part to the complexation of enzymes and antibodies.

Human Arginase 1 (hArg1) is an important therapeutic target for the treatment of various cancers, nervous system dysfunction, and cardiovascular dysfunction and diseases. While all published reports on inhibitors of hArg1 have focused on the discovery and optimization of small molecules, we have shown the use of antibodies to target and neutralize the function of hArg1. We have determined that the antibodies inhibit hArg1 without fully extending to, or engaging with, the catalytic Mn ions at the base of the active site, enabling the design of inhibitors that are not limited to the sterically narrow hArg1 active site. These antibodies represent compelling tools to asses, in physiologically relevant cell-based assays, their ability to block arginase mediated T-cell suppression, as has been shown for small molecule inhibitors. We intend these studies to be the subject of a future publication.

Beyond the scope of hArg1 specifically, the identification and characterization of highly selective inhibitory antibodies can be used as a discovery tool in the pursuit of enzymatic inhibitors that fall outside of the traditional methods of enzymatic inhibition including allosteric mechanisms, and substrate, transition state, or product mimicry. Designs inspired by such knowledge could include inhibitors—both small molecules and peptides—that bind at the surface of the enzyme and act through steric occlusion, regardless of how deeply, if at all, they enter the active site. In addition, inhibitors that bind outside of the active site but interfere with surface residues vital to enzymatic function allow for more freedom both in modification of chemical moieties on the inhibitor, and in conformational restrictions that would otherwise be present when trying to fit an inhibitor into the active site.

Undertaking studies to identify and refine antibodies as therapeutic treatments is a burgeoning strategy that will lead to future success in the treatment of diseases in the immuno-oncology space and beyond. When also considering the knowledge gained through use of inhibitory antibodies as probes for new inhibitor molecule design, the determination of antibody:enzyme structures is an important and worthwhile endeavor that will further the development of therapeutics in the future.

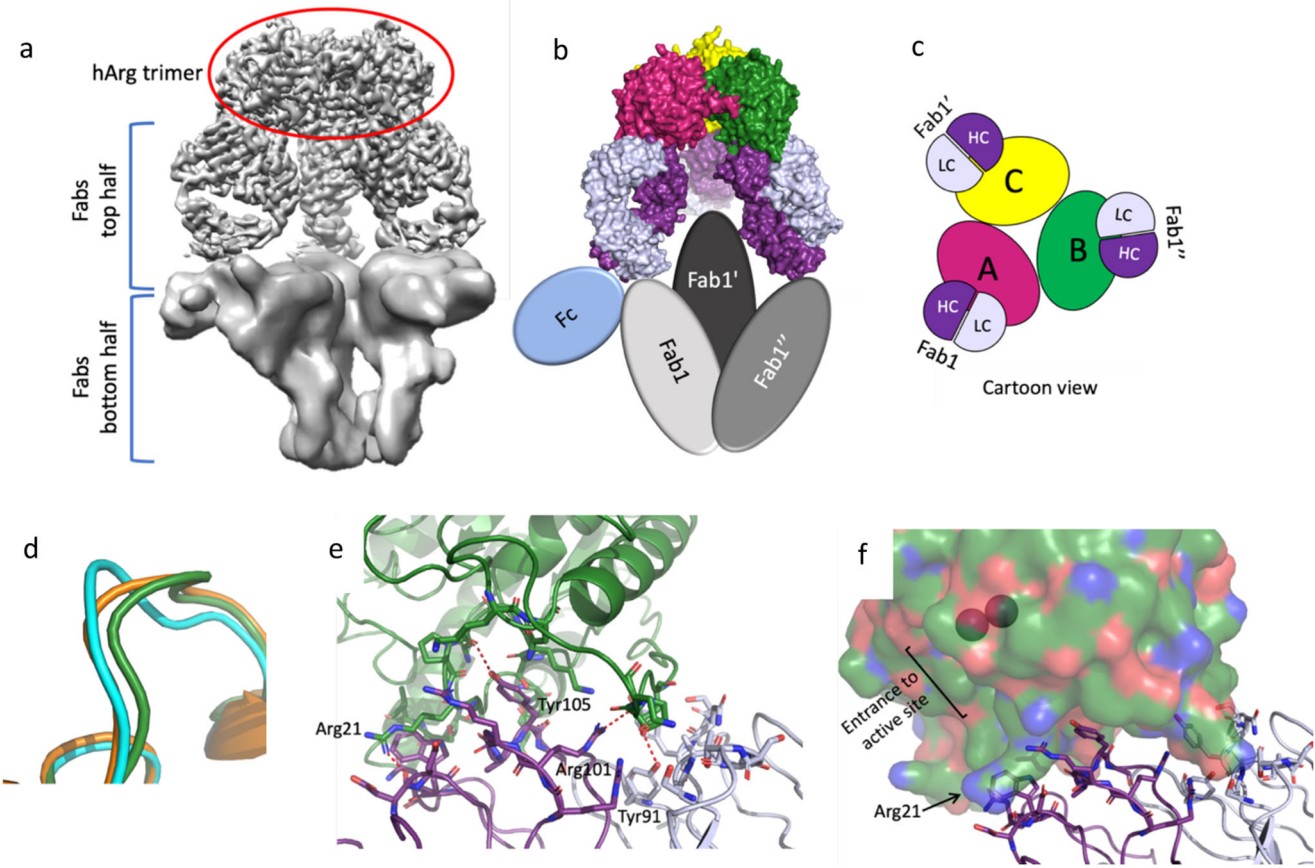

**Fig. 5 Overall complex formation and epitope mapping for mAb5 1:3 complex. a** Density for the bottom half of the complex is almost completely absent. In masked maps at very low contour it is possible to visualize some density for the other Fabs and the Fcs, but the Fabs on the bottom half appear to be in a different conformation and closer to each other with no density for Arginase trimer present. **b** This panel shows an overview of how the 1:3 hArg1 to mAb5 complex assembles, with the three monomers of the hArg1 trimer colored in green, pink, and yellow, and the mAbs colored in dark purple (heavy chain) and pale purple (light chain). The protein surfaces are shown for the top half of the complex, and the bottom half Fabs and one Fc shown in colored ovals. **c** A depiction of the complex for ease of interpretation is shown. Each antibody interacts with only one hArg1 monomer and these interactions are symmetric around the trimer. **d** The loop containing residues Lys16-Val24 is shown for hArg1 bound to mAb5 antibody (cyan), hArg1 bound to mAb1 (green), and hArg1 bound to a small molecule (orange) are depicted. **e** Details of the interactions between the Fab and the hArg1 monomer are shown, with several residues involved in hydrogen bonding interactions labeled. **f** A surface view of one of the hArg1 monomers (green) shows that the active site is fully exposed in this complex. The binuclear active site manganese ions are shown as purple spheres.

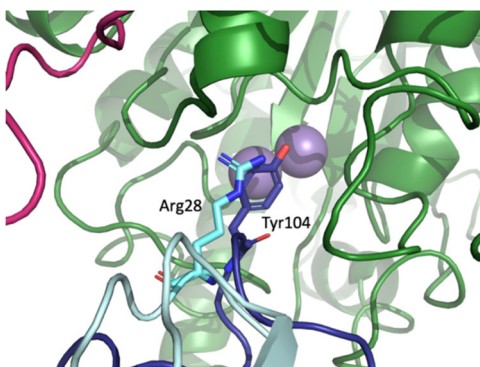

**Fig. 6 Comparison of mAbs 1-3 and mAb4 epitopes.** A closeup view of the hArg1 active site reveals the differences in epitopes between mAbs 1-3 and mAb4. The CDR-3 HC Tyr104 of mAbs1-3 (dark blue) is replaced by the CDR-1 LC Arg28 in mAb4 (cyan). In both cases the hArg1 active site is completely blocked by the antibody.

## Methods

**Expression and purification of hArg1**. Full-length untagged hArg1 was expressed in *E.coli* BL21 (DE3) using superbroth media. Expression was induced with 1 mM

IPTG at $OD_{600}$ 0.8 and cells were grown for 4 h at 37 °C. Cell pellets were resuspended in lysis buffer (10 mM Tris pH 7.5, 5 mM $MnCl_2$, 2 mM BME, 1 mg/ml lysozyme), passed through a microfluidizer three times at 15,000 PSI and the soluble fraction was clarified by centrifugation at $11,000 \times G$. Clarified lysates were heat treated at 60 °C for 20 min. Heat treated lysates were passed through a HiTRAP-SP column (GE). Flow through containing hArg1 was diluted to ~40 mM NaCl and reloaded on another HiTrap-SP column. hArg1 was eluted from the column using a linear gradient from 20 mM NaCl to 1 M NaCl. Pooled fractions were concentrated and loaded on a HiLoad Superdex 200 26/60 size exclusion column in 25 mM HEPES pH 7.3, 150 mM NaCl, 1 mM $MnCl_2$. Peak fractions were analyzed by SDS-PAGE, pooled and concentrated. *Purification adapted from Strickland Acta Cryst. (2011). F67, 90–93*

**Antibody discovery and optimization**. De novo antibody discovery for mAbs1-3 against arginase were executed on pre-immune yeast display libraries with a diversity of $10^{10}$ (Sivasubramanian et al., 2017). The soluble proteins used in the yeast display selections are biotinylated recombinant proteins. All proteins were analytically and verified by SEC and SDS-PAGE. Briefly, a yeast IgG library was subjected to multiple rounds of selection by magnetic and florescence activated cell sorting (BD ARIA III) in PBS buffer containing 1 mM $MnCl_2$. Selections were performed using 100 nM arginase followed by rounds of enrichment using decreased antigen concentrations to enhance for higher affinity binders. Top clones were isolated by affinity maturing its parental clone through shuffling the LC and optimizing HC CDR1 and CDR2 sequences. The selection of optimization libraries was repeated using 10 nM arginase and the isolated clones were then sequenced to identify the unique antibodies and screened for binding profiles by Octet Red. MAb 4 was identified in a campaign run at Precision Anitbody and mAb5 was identified

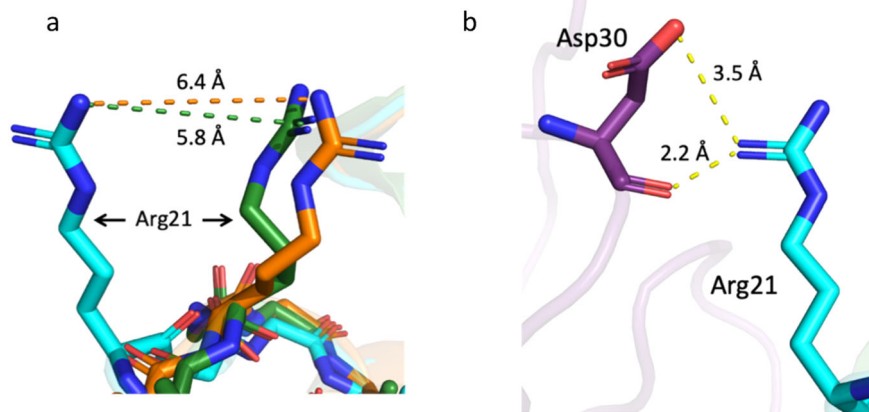

**Fig. 7 Comparison of an hArg1 loop when bound to mAb5, mAb1 and small molecule-bound hArg1.** An overlay of the hArg1 loop containing residues Lys16-Val24 is shown in Fig. 5d. When bound to the mAb5 antibody, the positioning of this loop (cyan) is altered when compared to hArg1 bound to mAb1 (green) or hArg1 bound to a small molecule (orange). **a** A closeup of Arg21 in all three structures highlights the difference in orientation of Arg21 which moves outward by 5.8–6.4 Å. **b** Arg21 interacts with Asp30 of the mAb5 heavy chain, which sits ~2.2–3.5 Å away.

via a Trianni mouse immunization campaign run at LakePharma. The antibodies were variable region sequenced and the variable heavy region was cloned into a human IgG1 encoded region of a pTT5 vector. The variable light region was cloned into a human kappa encoded region of a pTT5 vector.

**Antibody production**. ExpiCHO-S cells growing in suspension were transfected with indicated antibody expression plasmids (HC + LC) using commercially available protocols and ExpiFectamine CHO reagents (Thermo-Fisher)[50]. In brief, cells were transfected day 0 using 1 ug total DNA (3:2 ration LC:HC) per 1 ml cells at a density of 6 M cells per mL and a viability >95% measured using a Vi-Cell (Beckman-Coulter). On day 1, ExpiCHO feed and enhancer were added and culture temperature was lowered to 32 °C. On day 5, a second EXPI-CHO feed was performed and cell viability was measured using a Vi-Cell (Beckman-Coulter). Cultures were harvested between day 8 and day 12 depending on a cell viability >80%. Antibody was purified from clarified supernatant using Protein A chromatography (mAbSelect Sure LX, GE Healthcare). Protein A was incubated with the clarified supernatant overnight in 4 °C on a roller mixer. Resin was then collected from supernatant mixture and transferred into a column and washed with 10 column volumes (CV) of PBS. Elution of Mab was achieved using 20 mM sodium acetate, pH 3.5. 1CV fractions were collected and tested by Bradford assay to determine presence of protein. In some cases Protein A purification was followed by anion exchange chromatography (Capto Q, GE Healthcare). Purified antibodies were buffer exchanged into the final formulation buffer of 20 mM sodium acetate, 9% sucrose, pH 5.5. Purified antibody was checked for purity by reduced and non-reduced CE-SDS(Perkin-Elmer), concentration was measured by $A_{280}$, and aggregate content was analyzed by SEC-UPLC using a BEH200 UPLC-SEC analytical column (Waters corporation). Endotoxin was quantified using Endosafe® nexgen-MCS™(Charles River). Intact mass was confirmed via Synapt G2S QTOF or Xevo-TOF (Waters).

**Isothermal titration calorimetry assay**. Isothermal titration calorimetry (ITC) was conducted using a MicroCal PEAQ-ITC Automated (Malvern Inc, Westborough, MA) to determine the stoichiometry and thermodynamics properties of the hArg1:mAb interaction. Human Arg1 trimer and mAb4 (mouse human chimera IgG4 mAb) were prepared in 10 mM HEPES pH 7.4 buffer containing 150 mM NaCl and 1 mM MnCl₂. Arginase (100 uM) in the ITC syringe was titrated into the antibody (10 uM) in the ITC cell at 25 °C. Reference power was set to 10 ucal/sec with initial delay of 60 s and stirring speed of 750 rpm. Injection volume was 0.4 uL for the first injection and 4 uL for subsequent injections. Injection duration was 0.8 s for the first injection and 6 s for subsequent injections with 150 s spacing. Baseline was adjusted using buffer-buffer titration. Data analysis was done using MicroCal PEAQ-ITC Analysis Software. ITC results can be found in Supplementary Information Fig. 1.

**Size exclusion chromatography with multi-angle light scattering (SECMALS)**. SEC was performed on a Waters ACQUITY UPLC H-Class system equipped with a BEH 450 Å, 2.5 μm column (Waters). The sample was prepared by diluting to a final concentration of 1 mg/ml and injecting 20 μg. Isocratic flow of mobile phase buffer, 1X PBS, 0.02% sodium azide (pH 7.0) was run at 0.5 mL/min. The separation was conducted at ambient temperature and the column effluent was monitored at 280 nm. MALS analysis of the sample was performed continuously on the SEC column eluate, as it passed through a μDAWN MALS detector and an Optilab UT-rEX refractive index detector for UHPLC (both Wyatt Technology).

Data were analyzed with the ASTRA® software (Wyatt Technology). Results found in Supplementary Information Fig. 2.

**Cryo-electron microscopy methods and image processing**. All the Arginase: MAB complexes were formed by mixing the protein and the MAB in reaction buffer (25 mM HEPES pH 7.4, 150 mM NaCl, 1.0 mM MnCl₂) at 3:1 molar ratio and incubating the mixture for 30' before preparing the grids. Grids were prepared and data were collected at NanoImaging Services (San Diego, CA) according to the following protocol and the specifications in Tables 1 and 2: After incubation, the sample containing the complex was diluted with reaction buffer to ~75 uM concentration of monomeric Arginase, then mixed with DDM to the CMC concentration to reduce particle aggregation and used immediately afterwards to freeze grids. 3 ul of each sample were applied to 1.2/1.3 grids (Au/Au Quantifoil or C/Cu C-flat) which have been previously plasma-cleaned using a Gatan Solarus (Pleasanton, California) and mounted in a Vitrobot Mark IV. The sample was then blotted with filter paper for 6 s and plunged in liquid ethane. Electron microscopy was performed using a Thermo Fisher Titan Krios (Hillsboro, Oregon) transmission electron microscope operated at 300 kV and equipped with a Gatan Quantum 967 LS imaging filter and Gatan K2 Summit direct detector. Automated data-collection was carried out using Leginon software[51] in counting mode, collecting between 1500 and 3000 movies per sample at a defocus range between −1.0 and −2.0 μm, calibrated pixel size of 1.04 Å/pix and total dose of 45 e⁻/Å².

Movies were aligned using MotionCor2[52] and the CTF calculated using CTFFIND4[53] within the Appion package[54]. Aligned micrographs were imported into cryoSPARC[55] where all the subsequent steps of image processing were realized following a standard single particle workflow until the final reconstruction. Particle picking was performed based on blobs for mAb1. When the 2D classes from mAb1 became available, they were used as templates to select particles for the other samples. Several rounds of 2D classification and ab-initio were performed to select the best-looking particles and separate the different oligomerization complexes when they existed. In all the cases non-uniform refinement and higher-order CTF refinement were used to generate the best reconstructions. Symmetries C3 or C1 were enforced during the final refinement when applicable. When used during the refinement, masks were generated in Chimera[56]. Initial models for the structures were generated using MOE 2018.01 (Chemical Computing Group ULC, Montreal, QC, Canada) and the available structure of hArg (PDB ID = 6V7C)[20]. The structures were built in Coot[57] based on the cryo-EM density maps and subjected to one round of real space refinement in Phenix[58]. More information on cryo-EM methods and processing can be found in Supplementary Information Table 1 and Supplementary Information 3–5.

**Antibody potency and mechanism of hArg1 inhibition**. To evaluate antibody potency and mechanism of arginase inhibition, the antibodies to be tested were diluted in assay buffer (50 mM Tris pH 7.5, 50 mM sodium chloride, 1 mM Mn chloride and 0.05% bovine serum albumin) to a concentration 2.5-fold higher than desired assay concentrations. To each well of a Greiner black 384-well assay plate (catalog #781086) was added 10 μL of antibody solution followed by 10 μL of assay buffer or assay buffer with 1 nM human or mouse arginase. After 30 min of incubation at room temperature, 5 μL of a 5× solution of thioarginine (variable concentrations) was added. The assay was allowed to proceed for 60 min then quenched by addition of 15 μL of a solution of 375 μM uM 7-Diethylamine-3-(4-maleimidophenyl)−4-methylcoumarin (Sigma Chemical) in 70% ethanol was added to quench the reaction and detect thioornithine. The plate was briefly shaken to mix and the fluorescence was measured in an Spectramax plate reader

(Molecular Devices) with a 410 nm excitation wavelength and an 490 nm emission wavelength. Kinetic data were fit to various models of enzyme inhibition (competitive, mixed, noncompetitive and uncompetitive) using GraphPad Prism.

An alternate assessment of antibody potency was performed by serially diluting antibodies in assay buffer and performing the assay described above except that a fixed concentration of 1 mM thioarginine was used. Data were normalized to wells containing either no antibody (no effect control) or no arginine (max effect control). The percent inhibition was then fit to a four-parameter logistic equation in GraphPad Prism to determine the IC50 (graphs on next page). Graphs showing the competitive inhibition profile can be found in Supplementary Information Fig. 6.

**Determination of recombinant, human Arg1 activity by LC-MS.** Arginase enzyme was diluted to 1.25 nM in assay buffer containing 50 mM Tris pH 7.5, 50 mM NaCl, 1 mM MnCl$_2$ and 0.05% Bovine serum albumin and 20 μL were added to black 384 well assay plates (Greiner Bio-One Inc., Monroe, CA). Titrations of antibodies were added to appropriate wells on the plate, following which samples were incubated at 37 °C for 30 min in an oven. The arginine substrate was diluted in assay buffer to 1.5 mM and 5 μL were added into assay plate (final substrate concentration = 300 μM). The reaction was carried out for 1 h at 37 °C. At the end of the reaction, 15 μL of ethanol containing internal standards (40 μM [$^{13}C_6$]-Arginine and [$^{13}C_5$]-Ornithine) were added to the mixture to quench the reaction. The assay plate was then stored at −80 °C until sample derivatization.

10 μL of thawed sample were transferred from the assay plate to a 384 deep-well plateand 90 μL methanol were added. Samples were centrifuged at 2500 *rpm* for 10 min at room temperature and 5 μl of supernatant were transferred to a fresh 384 well Siliguard-coated plate containing 35 μL of neat borate buffer. AccQ·Tag Ultra reagent (Waters Corporation, Milford, MA) was initially prepared according to the manufacturer's instructions and then diluted an additional twofold with acetonitrile. 10 mL of the reconstituted AccQ·Tag were added to each well and plates were sealed with pierceable aluminum foil (Agilent Technologies, Santa Clara, CA). The derivatization reaction was carried out by incubating at 55 °C in an oven for 30 min. Plates were then stored at 4 °C prior to LC-MS analysis.

Samples were analyzed on a Thermo TSQ Vantage triple quadrupole mass spectrometer with an electrospray ionization source operating in the positive ion mode (Thermo Fisher Scientific; Waltham, MA). Separation was achieved using an Acquity UPLC HSS-T$_3$ 2.1 × 30 mm, 1.8 μM column (Waters Corporation, Milford, MA) at ambient temperature. A binary solvent system composed of 0.1% formic acid in H$_2$O (mobile phase A) and 0.1% formic acid in acetonitrile (mobile phase B) was used for chromatographic separation. Selected reaction monitoring was used to quantify the analytes and internal standards, with the specific precursor to product transitions of: Arginine (344.96 > 70.14), [$^{13}C_6$]Arginine (351.102 > 74.3), Ornithine (473.29 > 171.15), [$^{13}C_5$]Ornithine (478.29 > 171.15). During analysis samples were maintained at 10 °C in the autosampler chamber. For injection, 12 μL of sample were over-filled into a 5 μL sample loop. All results were analyzed using the response ratio of analyte peak area/internal standard peak area for data normalization. Dose response curves can be found in Supplementary Information Fig. 7.

**Affinity measurement of anti-ARG1 antibodies to human ARG1 by surface plasmon resonance.** The binding affinities of anti-ARG1 antibodies to monomeric and trimeric human ARG1 were measured by capturing human mouse chimeric antibodies on an anti-mouse IgG surface or human/humanized antibodies on an anti-human Fc surface on a Series S Sensor Chip CM5 (Cytiva) using a Biacore T200 or Biacore 4000 biosensor (Cytiva). Immobilization and affinity measurements were performed in 10 mM HEPES, 150 mM NaCl, 0.05% v/v Surfactant P20, 3 mM EDTA, pH 7.4 (Cytiva) at 25 °C. The anti-mouse IgG and anti-human Fc capture antibodies were immobilized on all surfaces of the chip following the Cytiva's Amine Coupling Kit, Mouse Antibody Capture Kit and Human Antibody Capture Kit protocols. To measure the affinity of each interaction, the anti-ARG1 antibodies were captured at 2 nM at 10 μL/min and 5–6 concentrations of a twofold dilution of human ARG1 from 100 nM were injected at 30 or 50 μL/min. A reference flow cell without captured antibody was also included. After the antigen injections, the dissociation of the interaction was monitored for 5 or 10 min. The antibody and antigen were removed from the chip with a 30 s injection of 10 mM glycine at pH 1.5 for anti-mouse IgG captures or 3 M MgCl$_2$ for anti-human Fc captures between binding cycles.

The data were processed and fit using Biacore T200 Evaluation Software version 2.0 or Biacore 4000 Evaluation Software version 1.1 (Cytiva). The data were "double referenced" by subtracting the response from the reference control flow cell and that from a buffer injection. The data were then fit with the '1:1 Binding' model to determine the association rate constant, $k_a$ (M$^{-1}$s$^{-1}$, where "M" equals molar and "s" equals seconds) and the dissociation rate constant, $k_d$ (s$^{-1}$). These rate constants were used to calculate the equilibrium dissociation constant, $K_D$ (M) = $k_d/k_a$. SPR data can be found in Supplementary Information Table 6.

**Molecular dynamics protocol.** Initial Cartesian coordinates for the Arg1 and mAb5 systems were generated from the 3.1 Å resolution crystal structure (**PDB: 7LEY; EMDB: 23294**). The system used for the simulation contained one monomer of Arg1 and the 2 mAb5 subunits (LC and the HC). The resulting structures were prepared using the leap module of the AMBER 16 package. The ternary system was solvated in a box of TIP3P[59] water molecules extending at least 10 Å beyond the protein and chloride ions were added to maintain charge neutrality. The ff15ipq force field[60] was used to construct the topology files for the protein. The initial structure was conjugate gradient (CG) minimized for 200 steps for the water molecules only, followed by 10,000 steps of CG optimization of the entire system to remove any bad contacts. After minimization, the full system was gradually heated from 0 to 300 K using a constant NVT ensemble for 100 ps of MD. The system was then switched to a constant NPT ensemble at 300 K and 1 atm using a coupling value of 2.0 ps for both temperature and pressure and ran for 100 ps. Following equilibration, 100 ns of production data were collected at constant NPT using the GPU-accelerated version of AMBER 16. Hydrogen bond analysis and distance calculations were computed using the CPPTRAJ program within AmberTools[61]. A chart showing results from the molecular dynamics simulation can be found in Supplementary Information Fig. 9.

**Reporting summary.** Further information on research design is available in the Nature Research Reporting Summary linked to this article.

## Data availability
The datasets generated during and/or analyzed during the current study are available. The structure factors and coordinates have been deposited with the Protein Data Bank and Electron Microscopy Data Bank as follows:

**mAb1** 2 hArg1:3 mAb1—PDB code 7LEX; EMDB code EMD-23293
**mAb1** 2 hArg1:2 mAb1—PDB code 7LEZ; EMDB code EMD-23295
**mAb2**—PDB code 7LF0; EMDB code EMD-23296
**mAb3**—PDB code 7LF1; EMDB code EMD-23297
**mAb4**—PDB code 7LF2; EMDB code EMD-23298
**mAb5**—PDB code 7LEY; EMDB code EMD-23294

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

## Acknowledgements

This work was supported by Merck Sharp & Dohme Corp., a subsidiary of Merck & Co., Inc., Kenilworth, NJ, USA.

## Author contributions

R.L.P, Y.G-L., S.G., C.S., G.S. and G.N.F. contributed to the acquisition, analysis, or interpretation of the structural data. V.J., M.A.B., L.F-D., M. Handa, E.K., X.C., J.R.M., N.N., J.O., H.S., K.C., S.T., D.C., M. Hsieh, and H.G. contributed to the acquisition, analysis or interpretation of the biological and biophysical data. V.J., L.F-D., B.H., M. Handa, E.S., A.S., H.L., S.T. and H.S. contributed to or directed the design and production of protein and/or antibodies. All authors critically reviewed or revised the paper for the intellectual content and approved the final version.

## Competing interests

Merck & Co., Inc., has filed provisional patent applications related to this paper. All authors are employees or former employees of Merck Sharp & Dohme Corp., a subsidiary of Merck & Co., Inc., Kenilworth, NJ, USA and may hold stock or stock options in Merck & Co., Inc., Kenilworth, NJ, USA.
