## [Peer Review File · Communications Biology]

Reviewers' comments:

Reviewer #1 (Remarks to the Author):

Arginase is a promising target for cancer immunotherapy by its regulation of T cell immunity in the tumor microenvironment, where it is secreted by myeloid cells. All published Arginase 1 inhibitors are small molecules and Palte et al. report the development of antibodies raised against human Arginase that may potentially be used as therapeutic antibodies. In view of the extracellular role of Arginase in regulating cancer immunotherapy, and the potential, unwanted toxic side effects of inhibiting intracellular Arginase, this is a valuable approach. The current manuscript describes the resolution and characterization of antibody - enzyme complexes. The Supplementary contains some activity data. However, these are only results of biochemical binding and enzyme assays. These results should be incorporated in the main manuscript. Furthermore, the activity of the antibody drugs should be studied in physiologically relevant cell-based assays, in particular, co-culture assays of relevant immune cell fractions. These assays may be performed with healthy donor materials, if the authors have no access to patient material isolated from the microenvironment of tumors.

Reviewer #2 (Remarks to the Author):

In this work, Palte et al. reported cryo-EM structures of five different mAbs in complex with human Arginase 1 (hArg1), which suggested that mAbs could inhibit the enzymatic activity of hArg1 through different mechanisms. hArg1 is an important target for the treatment of cancer and other diseases. Using mAb instead of small-molecule inhibitors to target hArg1 is a new and promising strategy. This work demonstrated that antibodies could be used to effectively inhibit enzymatic activity of hArg1. Similar strategy can be further explored and optimized for other enzymes. Moreover, structures of inhibitory mAbs bound to enzymatic proteins are valuable.

Overall the study is solid with cryo-EM structures and biophysical and biochemical characterizations (ITC, MS, SPR, SEC-MALS, etc). The manuscript was well written, but the structures were poorly presented with a lot of information omitted. Here are the major concerns:

1. The authors should follow other cryo-EM studies in the field to show at least a representative micrograph, 2D class averages, a workflow of data processing, a local resolution map, and FSC curves between half maps and model vs map for each dataset.
2. Whenever a map is shown, the threshold should be given.
3. The authors should consider showing the interfaces in figures instead of listing the interacting residues in the SI Tables which is far less intuitive.
4. Figure 1, the authors should consider coloring the map using the atomic models (instead of pointing out using circles). Both sharpened and unsharpened maps should be presented.
5. Figure 2, the panel letter A is covered (also for SI Figure 5). A cartoon representation may be more effective showing different monomers, the LC and HC for the zoomed view in the top right panel. An intermediate magnification is needed for panel C to show the position of CDR-3 and the hArg1.

Minor points:

1. SI Figure 1, what is TOGA? In line 125 to 127 of the manuscript, the authors said, "Data were fit to various models of inhibition including ...(SI Fig 1)". Is the fitting not shown?
2. Line 224, please change electron density to cryo-EM density. Technically a cryo-EM map is NOT electron density.
3. Fig. 5D and Fig. 7A are exactly the same, thus redundant.

Reviewer #3 (Remarks to the Author):

Human Arg1 is an essential target for the treatment of various diseases. They revealed the novel inhibition mechanism of human Arginase 1, based on the complex structure of Arg1 and monoclonal antibody. The author group identified and characterized five potent full-length anti-hArg1 antibodies. They revealed the full structural analysis of five large hArg1:mAb complexes by cryo-EM. Furthermore, the structures give us insights into how mAb inhibits the enzyme active site.

We can see the number of monoclonal antibody structures in PDB, but never see the whole mAb structures. In this meaning, the cryo-EM structure of the 650KDa complex is impressive. Significantly, they analyzed six structures using five different mAb 1~5, and the binding mode are different. The various experiments were done properly. I added some minor comments to improve the manuscripts.

Minor comments

(1)

Fig1

Fc was marked in one part of the middle figure. I think three Fc would be visible. Is this right? Please add two others.

This structure has two fold symmetry. Thus the upper and the lower Arg1 should be the same. However, the bottom hArg1 trimer seems flexible: B-factor is low. Why? left fig.

Typo: hArg1 bottom trimer?

(2)

Fig2 upper middle

Fc was marked in one part of the middle figure. Three Fc would be visible?

(3)

Fig4

Fc was marked in one part. Three Fc would be visible?

(4)

Fig5

Though the upper part are clearly visible, Fabs bottom half are disordered.

Why can these differences be seen?.

Was focused refinement applied on the upper parts?

Fc' and Fc' should be there?

(5)

In all structures, Fc parts were not visible. Please add an explanation of this.

(6)

Fig6

Arg28 on the LC and Tyr 104 on the HC are in the same active site of Arg1. These characters of two amino acids are different, so it is not usual as they occupied the same position.

The broad area by other amino of mAb contributes the binding mainly, and these amino acid contributions are not large. How about the binding contribution of these amino acids? Please add comments and discussions.

(7)

In this paper, the author showed the whole mAb complex structure. It is interesting in science, but is it important for drug development?

(8)

In SI Table 7, please add refinements parameters.

Model composition

Non-hydrogen atoms (Arg1 and mAb)

Protein residues (Arg1 and mAb)

Ligands

B factors (\AA^2)

Protein (Arg1 and mAb)

Ligand

*** Please note: the entire SI has been reconfigured in order to more correctly follow the introduction of material in the manuscript. This was a personal decision and is unrelated to reviewer comments. The responses below refer to the new SI numbering, which will not line up with the referee's comments so that they remain unaltered.**

Reviewer 1

Referee Comment	Response
Arginase is a promising target for cancer immunotherapy by its regulation of T cell immunity in the tumor microenvironment, where it is secreted by myeloid cells. All published Arginase 1 inhibitors are small molecules and Palte et al. report the development of antibodies raised against human Arginase that may potentially be used as therapeutic antibodies. In view of the extracellular role of Arginase in regulating cancer immunotherapy, and the potential, unwanted toxic side effects of inhibiting intracellular Arginase, this is a valuable approach. The current manuscript describes the resolution and characterization of antibody - enzyme complexes. The Supplementary contains some activity data. However, these are only results of biochemical binding and enzyme assays. These results should be incorporated in the main manuscript. Furthermore, the activity of the antibody drugs should be studied in physiologically relevant cell-based assays, in particular, co-culture assays of relevant immune cell fractions. These assays may be performed with healthy donor materials, if the authors have no access to patient material isolated from the microenvironment of tumors.	We understand the desire of the reviewer to have additional data beyond the discussed biochemical binding and enzyme assays. However, this project was deprioritized, and we do not currently have resources to conduct this work. Additional internal data that is focused on the biology of this target is currently being collated. We hope the information contained within this paper enables others to take up the mantle. To this end, we have added a statement in the manuscript conclusion section in lines 365-368: “These antibodies represent compelling tools to test in physiologically relevant cell-based assays to assess their ability to block arginase mediated T-cell suppression as has been shown for small molecule inhibitors. We intend these studies to be the subject of a future publication.”

Reviewer 2

Referee Comment	Response
1. The authors should follow other cryo-EM studies in the field to show at least a representative micrograph, 2D class averages, a workflow of data processing, a local resolution map, and FSC curves between half maps and model vs map for each dataset.	SI Figures 3, 4, and 5 have been added to the Supp. Info. to cover these aspects, and figure legends have been written for each. A sentence within the main manuscript has been added to direct readers to those figures. Lines 107-109 : Figures showing a representative micrograph, 2D classes, a flow chart of the main processing steps, and Gold-Standard Fourier-shell correlation curves for all structures can be found in SI Figures 3-5.
2. Whenever a map is shown, the threshold should be given.	Threshold values have been added to Fig 1 , which is the only figure showing a map.
3. The authors should consider showing the interfaces in figures instead of listing the interacting residues in the SI Tables which is far less intuitive.	We agree that the interactions can be hard to visualize by using the Tables alone. To this end, we have rearranged a few images within Figure 2 and added an image showing an enlargement of one of the interfaces. However, we believe that visually displaying all of the interactions across the extended surface interface would result in very crowded, and likely indecipherable, images. We intend for this image to serve as a representation of the types of interactions – salt bridges and H-bonds – that drive the binding and potency of the anti-hArg antibodies. The manuscript was updated to address this, and the following statement was added in lines 163-165: A close up view of one hArg1:mAB1 interface is shown in Figure 2B as a representative image of the numerous surface interactions found between hArg1 and mAbs.
4. Figure 1, the authors should consider coloring the map using the atomic models (instead of pointing out using circles). Both sharpened and unsharpened maps should be presented.	This has now been addressed in Figure 1, including an updated figure legend as seen in Figure 1 at the end of this document.

5. Figure 2, the panel letter A is covered (also for SI Figure 5). A cartoon representation may be more effective showing different monomers, the LC and HC for the zoomed view in the top right panel. An intermediate magnification is needed for panel C to show the position of CDR-3 and the hArg1.	We thank you for catching these. The labels have been corrected. We appreciate the push for clarity. The image has been updated by adding a simplified cartoon representation and as is shown in Fig2B. The figure legend has been updated to reflect this. An intermediate magnification image that shows the full extended loop has been added to Figure 2, panel D. Figure legend has been updated to reflect this.
6. SI Figure 1, what is TOGA?	The definition of TOGA has been added to SI Figure 7 legend, line 465: “SI Figure 7. Dose response curves as determined by LCMS (mAb1 and mAb2) and TOGA (ThioOrnithine Generation Assay) (mAb5).”
7. In line 125 to 127 of the manuscript, the authors said, “Data were fit to various models of inhibition including ...(SI Fig 1)”. Is the fitting not shown?	The fitting is shown in SI Fig 6, but we appreciate that it may not be intuitive and/or completely explained. We have added the following statement to the manuscript, lines 131-137, to better reflect this: “Data were fit to various models of inhibition including competitive, mixed, noncompetitive, and uncompetitive and the quality of fit was evaluated using the Aikakie information criteria³⁶. (SI Fig 6). In each case the inhibition data was best fit by a competitive inhibition model and the fitted data are shown in SI Fig 6. Inferior fits of the data to other models of inhibition are not shown. The Ki for mAb1 and mAb2 was 3.3+/-0.3 nM and 5.3+/-0.8 nM respectively. The Ki of mAb5 against mouse arginase was 25+/-1.8 nM.”
8. Line 224, please change electron density to cryo-EM density. Technically a cryo-EM map is NOT electron density.	The reviewer is correct, and the text has been revised accordingly.

9. Fig. 5D and Fig. 7A are exactly the same, thus redundant.	We appreciate the comment. The redundant figure has been removed from Figure 7, and the figure legend text was slightly changed to refer back to Figure 5D. The two remaining images were relabeled as A and B and the manuscript text was updated to reflect this.
---	--

Reviewer 3

Referee Comments	Response
1. Fig1 Fc was marked in one part of the middle figure. I think three Fc would be visible. Is this right? Please add two others.	We do understand that the lack of the second and third Fcs can be misleading. However, with the generally trimeric nature of the structures, the Fcs prove difficult to represent while maintaining image clarity. One of the other two Fcs would either point out behind the image and therefore be unseen or would be modeled in the front of the image, covering a large portion of the structures. Because of this desire to keep figures clean and clear we chose to only show one Fc on each structure included in this manuscript. We did add a sentence in the manuscript to further clarify this, lines 152-153: “While all three Fcs are present, for clarity only a single one is displayed in the image.”
2. This structure has two fold symmetry. Thus the upper and the lower Arg1 should be the same. However, the bottom hArg1 trimer seems flexible: B-factor is low. Why?	We believe the reviewer is referring to low resolution in the bottom half, not low B-factor; higher B-factor relates to higher flexibility while a lower B-factor corresponds to more rigidity. We will respond based on that assumption. While the interactions between hArg:mAbs are identical in the top half compared to the bottom half, there is not perfect two-fold symmetry between the two halves. Rather, the mAbs have a slight twist, preventing two-fold symmetry from being applied. Because of this, we chose to focus on and refine around only one half the complex, allowing for high-enough resolution to view specific

	interactions between epitope and paratope. This was done with the sacrifice of resolution on the other half. The paper has been edited to better clarify this point in lines 115-121: Within each 2:3 complex the angle of each of the three mAb backbones is slightly different and therefore no perfect 2-fold symmetry could be applied between top and bottom halves of the complex during map reconstructions. This knowledge, in addition to the fact that the interactions between hArg1 and mAbs1-4 are identical for both halves, led to refinements focused on one-half of the complexes. This allowed for an increase in resolution of one half at the expense of resolution on the second half.
3. Typo: hArg1 bottom trimer?	Thank you for catching this. The figure has been corrected.
4. Fig2 upper middle - Fc was marked in one part of the middle figure. Three Fc would be visible?	Please refer to response for Figure 1 as to why the three Fcs are omitted for clarity.
5. Fig4 - Fc was marked in one part. Three Fc would be visible?	Please refer to response for Figure 1 as to why the three Fcs are omitted for clarity.
6. Fig5 Though the upper part are clearly visible, Fabs bottom half are disordered. Why can these differences be seen?	Even when processed at low contour, no second hArg1 trimer could be seen. Because of this, we believe the density is likely poor due to flexibility of the lower half of the antibody arms since they are not held in place by binding interactions with hArg. This flexibility leads to disorder within the cryoEM map as there is no consistent orientation of the lower Fabs. To help clarify this within the manuscript we have added lines 233-237: “Lacking this second hArg1 trimer, the “bottom” set of Fabs are no longer conformationally restricted through binding interactions with hArg1. This flexibility correlates with the absence of clear density for the entire second half of the complex.”

Was focused refinement applied on the upper parts? Fc' and Fc' should be there?	Focused refinement was not applied, as far as “focused refinement” is currently defined in Cryosparc. A C3 symmetry – due to the trimeric hArg1 – was applied which smeared out the poor density. Please refer to response for Figure 1 as to why the three Fcs are omitted for clarity.
7. In all structures, Fc parts were not visible. Please add an explanation of this.	An explanation was provided in lines 103-107: “The Fc (Fragment, crystallizable) regions of the antibodies do not interact with the hArg1 trimers and are only visible in unfiltered, unflattened maps at a very low contour (Figure 1). While the full mAbs were indeed part of the visualized complexes we have chosen not to show the Fc regions in the finalized structures due to the inability for clear placement and interpretation at an atomic level.” We hope this suffices to explain the purposeful omission of Fcs in all structures.
8. Fig6 Arg28 on the LC and Tyr 104 on the HC are in the same active site of Arg1. These characters of two amino acids are different, so it is not usual as they occupied the same position. The broad area by other amino of mAb contributes the binding mainly, and these amino acid contributions are not large. How about the binding contribution of these amino acids? Please add comments and discussions.	We agree that the shared broad surfaces contribute to significant interactions. Indeed, the surface area between each monomer and the hArg1 trimer is nearly 1400 Å². When this is triplicated across the trimer the overall surface area is approximately 4200 Å². A significant amount of these interactions are a compilation of hydrogen bonds and/or salt bridges; the interactions include both side chain functional groups and backbone nitrogen atoms and carbonyl oxygen atoms. We were intrigued to discover that both an Arg and a Tyr can occupy the active site tunnel and thereby directly occlude binding of arginine which is a different mechanism than mAb5 inhibition of hArg1. We do acknowledge that there are many more

	interactions taking place that contribute to the inhibition. The following has been added to the manuscript: Lines 301-303 - The shared surface area between a single hArg1 trimer and three Fabs is ~4200 A² and the interactions across these surfaces clearly contributes significantly to the overall potency of the mAbs. And lines 316-319 - The side chains of arginine and tyrosine are chemically different in size and charge, so it seems that, rather than relying on a specific residue for inhibition, overall steric occlusion of the active site is the inhibitory mechanism of the antibodies.
9. In this paper, the author showed the whole mAb complex structure. It is interesting in science, but is it important for drug development?	To our knowledge this is the first publication to show inhibition of an enzyme with a full-length mAb. While the majority of drugs on the market and currently in development are small molecules, there is growing interest in exploring other means of inhibition including Fabs, nanobodies, and scFvs, especially within the immunology field. Because of this, we believe that such studies and results are important milestones within the general drug development field.
10. In SI Table 7, please add refinements parameters.	GSFSC half-maps (A) and 0.5 FSC model-map (A) parameters have been added to SI Table 2.

Figure 1

Figure 1. Simple depiction and cryo-electron microscopy maps of the 2:3 hArg1:mAb1 complex. Far Left: Most complexes presented in this paper consist of two hArg1 trimers on the top and bottom of the complex spanned by three full mAbs, giving the complex a sandwich-like appearance. **Left:** An exemplar map for the hArg1:antibody structures described in this paper, colored to match the first image. **Right:** A color shell based on local resolution is overlaid with an unsharpened map. It is apparent that the more highly-resolved top half of the complex is almost completely within the 3-4 Å range, it is likely that the bottom half is just as ordered, but is flexible relative to the top half resulting in the seeming loss of resolution. The top half of each complex was used to map epitope and paratope interactions. **Far Right:** A sharpened map of the complex.

Figure 2

Figure 2. Epitope determination for mAb1, mAb2 and mAb3. A) This panel shows an overview of how the large 2:3 hArg1 to mAb1 complex assembles. The three monomers of the hArg1 trimer colored in green, pink, and yellow, and the mAbs colored in dark blue (heavy chain) and light blue (light chain). **Left:** Surface representation of the full complexes. **Middle:** A surface representation provides a closeup view of mAb1's interaction across two hArg1 monomers. **Right:** A closeup view of the mAb1:hArg1 monomers in simplified cartoon representation. B) The complex is viewed in a simplified cartoon form as viewed from the top (**left**). Each HC interacts with two hArg1 monomers while each LC interacts with only one hArg1 monomer; these interactions are symmetric around the trimer. **Right:** a closeup view of the monomerA:HC surface interaction. C) A sample of the electron density at the hArg1:Fab interface is shown, highlighting the ability to confidently model all main chain and side chain atoms. D) This class of mAbs are characterized by a very long CDR-3 loop (orange). Tyr104 (shown as sticks) extends into the hArg1 active site. The binuclear active site manganese ions are shown as purple spheres.

SI Figure 3

SI Figure 4

SI Figure 4. Summary of main processing steps for the cryo-EM datasets to generate each of the final reconstructions. The number of particles (in thousands) involved in every step of the processing is indicated in blue italic letters for the steps of particle extraction, particle selection after 2D classification, and 3D refinement. The number of classes for 3D classification step (either by heterogenous refinement or ab-initio) is indicated in red letters when necessary. Global CTF refinement successfully improved the final resolution in the cases of the “mAb1 full complex”, “mAb1 masked” and “mAb5 1 trimer:3 Fabs” reconstructions. The map for the “mAb2 full complex” is represented at a lower threshold, for clarity purposes. As a consequence, the second trimer of the map is not displayed in this representation. Additionally, some repetitive steps of the processing have been omitted from this summary for clarity purposes.

SI Figure 5

SI Figure 6

388
389
390
391
392
393
394
395
396
397
398
399
400
401
402
403
404
405
406
407
408
409
410
411

SI Figure 6. Graphs showing the competitive inhibition profile of mAb1 and mAb2 against human Arg1 and mAb5 against mouse Arg1.

SI Figure 7

SI Figure 7. Dose response curves as determined by LCMS (mAb1 and mAb2) and TOGA (ThioOrnithine Generation Assay) (mAb5).

REVIEWERS' COMMENTS:

Reviewer #2 (Remarks to the Author):

The authors have addressed all my concerns. The only remaining issue is that there should be a color code or legend for the very right panel of figure 1 similar to that of figure 2.

Reviewer #3 (Remarks to the Author):

The manuscript was revised adequately. I confirmed that the author displayed only the first molecule for the clarity. And I understood the reason why B-factor is different between upper and lower.